# Effectiveness of the Booster of SARS-CoV-2 Vaccine among Japanese Adolescents: A Cohort Study

**DOI:** 10.3390/vaccines10111914

**Published:** 2022-11-12

**Authors:** Yoshika Saito, Kana Yamamoto, Morihito Takita, Masahiro Kami, Masaharu Tsubokura, Kenji Shibuya

**Affiliations:** 1Research Division, Medical Governance Research Institute, 2-12-13-201 Takanawa, Tokyo 108-0074, Japan; 2Department of Radiation Health Management, Fukushima Medical University, 1 Hikarigaoka, Fukushima 960-1247, Japan; 3Soma COVID Vaccination Medical Center, 63-3 Nakamura-Kitamachi, Soma 976-8601, Japan; 4Tokyo Foundation for Policy Research, 3-2-1 Roppongi, Tokyo 106-6234, Japan

**Keywords:** COVID-19, adolescent, vaccination, booster, infection

## Abstract

Vaccination is effective in preventing COVID-19-related hospitalization among all age groups, but there is limited evidence on the effectiveness of the booster of the SARS-CoV-2 vaccine among adolescents. We analyzed the data on the status of SARS-CoV-2 infection and their vaccination profiles in adolescents aged 13–18 years in Soma city (Fukushima, Japan) (*n* = 1835) from 14 May to 15 June 2022. The crude incidence rate and 95% confidence interval were calculated with the negative-binomial regression model after classifying the immunization status. The crude effectiveness of a booster administration to prevent infections was estimated as 86.4% (95% confidence interval: 57.2–95.7) when compared with the primary vaccination alone. The results of this study support that the community-based mass vaccination campaign of a booster dose among adolescents has additional protection from COVID-19 during the period of the B.1.1.529 (omicron) variant wave.

## 1. Introduction

Adolescents tend to exhibit milder symptoms of the coronavirus disease-2019 (COVID-19) compared with adults [1]; however, a surge in COVID-19 infections in adolescents can exacerbate the outbreak to the whole population [2], and the long-COVID sequelae such as tiredness, headache, and shortness of breath are major concerns to adolescents’ health [3,4]. Vaccination is effective in preventing COVID-19-related hospitalization among all age groups, which supports the role of the severe acute respiratory syndrome coronavirus 2 (SARS-CoV-2) vaccination in adolescents [5]. Of note, the vaccine effectiveness at addressing long-COVID symptoms in adolescents has not been evident yet [6,7]. The vaccine effectiveness is waning owing to both the time-dependent property of immunogenicity and the immune escape capacity of the variant of concern [8]. To date, there is limited evidence on the effectiveness of the booster of the SARS-CoV-2 vaccine among adolescents.

The City of Soma, Fukushima, Japan, implemented a mass vaccination campaign of the booster dose of BNT162b2 mRNA vaccine (Pfizer-BioNTech) among adolescents aged 13–18 years from 16 April to 7 May 2022. In collaboration with the Fukushima Prefecture, Soma City collected information on infected cases for public health purposes. The community-based mass vaccination is a public measure arising from lessons learned during disaster relief efforts for the Great East Japan earthquake, followed by the nuclear power plant accident in 2011 [9]. By employing an observational and descriptive analysis, we report the vaccine effectiveness of the booster among adolescents in this area.

## 2. Materials and Methods

### 2.1. Study Procedures and Ethical Considerations

The analysis in this study was performed using official data reported to the Medical Center for SARS-CoV-2 vaccination of Soma City (Fukushima Prefecture, Japan). The city has maintained vaccination records of its residents under the Immunization Act of Japan [10]. All COVID-19 cases in Japan have been reported to the public health office of prefectures under the Infectious Disease Control Law. Soma city and Fukushima Prefecture have performed collaborative efforts in monitoring COVID-19 patients. Our study group received the descriptive statistics on the SARS-CoV-2 vaccinations and COVID-19 cases from Soma City. This study was ethically approved by the Institutional Review Board of the Medical Governance Research Institute (Minato ward, Tokyo, Japan) (Approval Number: MG2022-04).

### 2.2. Study Population and Mass Vaccination

We received the descriptive statistics on immunization status on 7 May 2022, in all adolescents aged 13–18 years in the city and their COVID-19 infection between 14 May and 15 June 2022.

Adolescents aged 13–18 years in Soma City were given the opportunity of mass vaccination for the primary immunization with two doses of BNT162b2 mRNA vaccine (Pfizer-BioNTech) in July–August 2021 and a booster administration with BNT162b2 between 16 April and 7 May 2022. The SARS-CoV-2 vaccination was individually given if they missed the opportunity of the mass vaccination.

### 2.3. Diagnosis of SARS-CoV-2 Infecion

All cases diagnosed with SARS-CoV-2 infections in Japan must be reported to the prefectural health offices by the Infectious Disease Control Law by 25 September 2022. The diagnosis of infection was made based on the positive results of polymerase-chain-reaction (PCR) testing or rapid antigen testing. Physicians can diagnose COVID-19 if the individuals without any SARS-CoV-2 testing show cold symptoms and had close contact with known COVID-19 patients. The severity of COVID-19 was evaluated in accordance with the guidance of the Japanese Ministry of Health, Labour, and Welfare, which includes the common conditions with the National Institute of Health (NIH) guideline [11].

### 2.4. Statistical Analysis

Descriptive statistics were determined in Soma City for the vaccine status classified by the infection status of SARS-CoV-2. We employed the negative-binomial regression model to calculate the crude incidence rate and the 95% confidence interval (CI) with the group-level counted data, as over-dispersion was observed in the data [12,13]. The vaccine effectiveness was obtained as 1 minus the odds ratio of the booster group in COVID-19 cases compared with the primary vaccination alone group [14]. The data analyses were carried out with IBM SPSS software, version 28 (IBM Corp., Armonk, NY, USA). The study followed the STROBE reporting guideline.

## 3. Results

### 3.1. Study Population

In total, 1558 (84.9%) adolescents completed the primary vaccination, among which 1128 (61.5%) received a booster dose before the observation period (Figure 1).

### 3.2. Cohorts with SARS-CoV-2 Infection

No hospitalization related to COVID-19 was reported during the observation period. Seventeen patients with mild symptoms and one asymptomatic case were observed (Table 1). Moreover, 11 and 4 cases among 430 adolescents with primary vaccination alone and 1128 with a booster administration were observed, respectively. During the observation period, more than 99% of sequenced COVID-19 cases in Japan were omicron variants [15,16].

### 3.3. Crude Effectiveness of a Booster Administration

The crude effectiveness of a booster administration to prevent infections was estimated as 86.4% when compared with the primary vaccination alone (Table 2).

## 4. Discussion

This observational study suggests additional protection from COVID-19 by administering a booster dose among adolescents in Soma City (Fukushima Prefecture, Japan). A randomized clinical trial including adolescents demonstrated the effectiveness of a booster vaccination during the B.1.617.2 (delta) variant period [17]. The results here support the role of a booster vaccination for adolescents in COVID-19 prevention measures during the period of the omicron variant wave; however, the limitations of the present study should be carefully considered owing to the nature of the observational study and a short follow-up period.

We show here 86.4% (95% CI: 57.2–95.7) of vaccine effectiveness of a booster administration against SARS-CoV-2 infection for a month when compared with the primary-immunization-only population. Reports on the vaccine effectiveness of the booster in the omicron-variant period in adolescents are limited. A Singapore group investigated the vaccine booster effectiveness in adolescents aged 12–17 years: 56% (53–58) and 94% (86–97) against SARS-CoV-2 infection and hospitalization, respectively, in the omicron variant period, compared with those unvaccinated [18]. A U.S. study estimated 71.1% (65.5–75.7) of booster effectiveness against infection compared with those unvaccinated [19]. The observation period in both studies was up to two months. We do not compare our results to these studies because of differences in the control group, cohort size, and the trend of COVID incidence, as well as higher coverage of the primary vaccination in the Soma cohort. However, the results in every study suggest the usefulness of booster administration in adolescents in the omicron era.

The community-based mass vaccination of a booster dose has an advantage not only in the broad coverage of immunization, but in estimating its real-world effectiveness. Validating its effectiveness with real-world data will be of more importance as the updated bivalent vaccines were authorized with the immunogenicity data, not with clinical data [20]. The database where the infection records were linked with individual vaccination records is necessary to develop the future strategy for SARS-CoV-2 immunization [21]. Japan maintained universal health coverage; however, the vaccination and infection records were separately stored by municipalities and by the public health office of the prefectural government, respectively, owing to differences in the regulation. Such a difference in the responsible body of data storage makes it difficult to scientifically analyze real-world vaccine effectiveness. To the best of our knowledge, no reports on the vaccine effectiveness of a booster have been published in the Japanese adolescent population. An adult study in Miyagi Prefecture, Japan, analyzed the vaccine booster effectiveness in the omicron era; however, the study was conducted in outpatient clinics with the individuals with close contact to COVID-19 patients [22]. Of note, the Miyagi study supports the third dose of the SARS-CoV-2 vaccination to prevent infection as having 40% (95%CI: 20–60%) effectiveness, which was the highest at 10–70 days after receiving the third dose. The nationwide cohort studies such as in the United Kingdom [23], Denmark [24], Singapore [25], and Qatar [26] have demonstrated the benefits of a large database in understanding population-level vaccine effectiveness.

The City of Soma immediately planned community-based mass vaccination for the primary series and the booster dose of SARS-CoV-2 vaccination based on lessons learned from evacuation after the Great East Japan Earthquake [9]. The evacuation experience enhanced communication within intra-city areas, which enabled the city officers to organize the vaccination schedule by area quickly. The city has published the incidence of COVID-19 patients on a daily basis on the internet homepage [27].

Concern about the SARS-CoV-2 vaccination for adolescents includes adverse events such as perimyocarditis [28,29], which might cause vaccine hesitancy [30]. Soma city conducted recipient-oriented safety surveillance after the primary vaccination of SARS-CoV-2 and reported the results, which are publicly available for adolescents [31]. Safety surveillance is commonly designed at the national level to determine the rate, but for serious events, a large database is used [32]. A small study like the survey in Soma City is not enough to find unusual events. It is, however, beneficial to obtain the trust of residents by maintaining transparency in the influence of vaccination in the community. Soma city is the fastest municipality for the coverage of COVID vaccination, based on lessons learned from the disaster of the Great East Japan Earthquake in 2011 [33]. The city office balanced the vaccine policy between boosting vaccination with a community-based approach and ensuring safety by surveying and reporting adverse events, especially for adolescents. Misinformation that may cause vaccine hesitancy has been wildly spread on internet websites [34]. A detailed assessment is necessary to determine the relationship between their perception of vaccination and the characteristics of websites that they commonly refer for acquiring knowledge of vaccination.

There are some limitations in this study. A small cohort size and the nature of the observational study cause confounding bias. The study area indicated high coverage of the primary vaccination (84.9% before mass vaccination of booster in the study cohort), which might lead to an overestimation of the vaccine effectiveness owing to care-seeking behavior or healthy vaccine bias [35]. Diagnostic bias, where the physician may test more unvaccinated patients, might also influence the elevation of vaccine effectiveness [14]. False-negative test results of PCR or antigen tests cause misclassification [36,37]. We could not identify patients infected before the observation period, which can also contribute to bias [38]. The short observation period might influence the biases due to misclassification and prior infection. The clinical diagnosis of COVID-19 without the SARS-CoV-2 testing affects the reliability of the finding of this study; however, the no-test diagnosis was very limited. Twenty-three (0.3%) cases among 7353 COVID patients in the Fukushima Prefecture during the study period were diagnosed as COVID-19 without testing owing to difficulty in having the opportunity to take the test [39]. Lastly, the short observation period in this study also limits the generalization of this study, as the waning of vaccine effectiveness was reported [19].

## 5. Conclusions

This study demonstrated the vaccine effectiveness of a booster administration of the SARS-CoV-2 vaccine in adolescents aged 13–18 years in a city that performed mass vaccination in Japan when compared with those with primary administration alone, which supports the protection of SARS-CoV-2 infection during the omicron variant wave. The study results, however, are carefully generalized as there are limitations due to the small cohort, the nature of the observational study, and the short period of observation.

## Figures and Tables

**Figure 1 vaccines-10-01914-f001:**
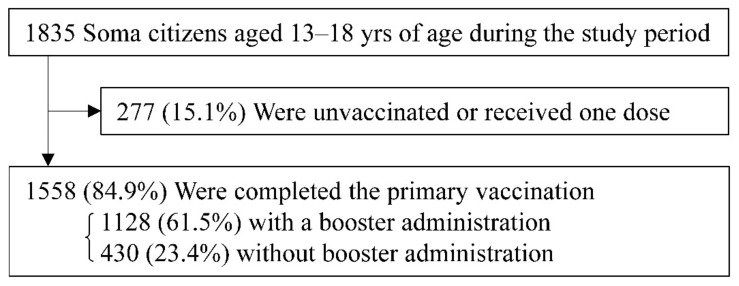
Immunization status of study population. The study population is shown for the analysis of the effectiveness of a booster administration of SARS-CoV-2 vaccination for adolescents in Soma, Fukushima, Japan.

**Table 1 vaccines-10-01914-t001:** Diagnosed COVID-19 cases classified by the status of SARS-CoV-2 vaccination in Soma adolescents (n = 1835).

		Vaccination Status	
	Unvaccinated or Partially Vaccinated for Primary Dose	Primary Vaccination Alone	A Booster Vaccination Received
	(*n* = 277, 15.1%)	(*n* = 430, 23.4%)	(*n* = 1128, 61.5%)
SARS-CoV-2 infection			
Asymptomatic	0	0	1
Mild symptom	3	11	3

The number of confirmed COVID-19 cases is shown by the status of SARS-CoV-2 vaccination.

**Table 2 vaccines-10-01914-t002:** Relative effectiveness of a booster administration of SARS-CoV-2 vaccination compared with primacy series vaccination alone in adolescents.

Vaccination Status	Number of Adolescents	Number of COVID-19 Cases	Crude Incidence Rate (95% CI)	Crude Vaccine Effectiveness (95% CI)
Primary vaccination alone	430	11	2.6 (1.4–4.6)	Ref
A booster administration received	1128	4	0.4 (0.1–0.9)	86.4 (57.2–95.7)

Abbreviation: CI: confidence interval.

## Data Availability

The descriptive data were provided from Soma city and are shown in Table 1. Data at individual citizen level, however, are not publicly available because of privacy restrictions in the government.

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
