# Peer review of "Effectiveness of the Booster of SARS-CoV-2 Vaccine among Japanese Adolescents: A Cohort Study"

_vaccines, 2022, doi:10.3390/vaccines10111914_

Round 1

Reviewer 1 Report

This communication aimed to examine the effectiveness of the SARS-Cov_2 booster administration among Japanese adolescents. The topic is very important; however, I have several concerns.

First, I am confused about the differences between the effectiveness of boosters and the effectiveness of booster administration. As I read the title, I think booster administration may indicate the level of adolescents to have a booster, but it was actually to examine the effect of infection. I think the effectiveness of the booster may be more appropriate.

Second, it will be better if the authors could provide more descriptions of the regression. What is the rationale to use the Poisson regression? Did the data meet the assumption of the Poisson regression?

Finally, it is good that the authors made several international comparisons. It will better if the authors could introduce Japanese experiences on both primary and booster vaccines to improve adolescent intake. 

Author Response

November 6, 2022

Dr. Prof. Dr. Ralph A. Tripp,

Editor-in-chief,

Vaccines,

Dear Dr. Tripp,

I am writing to submit a revision of our manuscript titled “Effectiveness of the booster of SARS-CoV-2 vaccine among Japanese adolescents: A cohort study” (Manuscript ID: vaccines-1990162). We sincerely appreciate the helpful comments from the reviewers and have revised our manuscript according to their suggestions.

The changes we have made to our manuscript are recorded with the track and change service. Please find below the detailed list of changes made in response to the reviewers' comments, along with an explanation for these changes.

We hope that the revised manuscript will now be acceptable for publication in the Vaccines and look forward to hearing your decision. Thank you very much for your consideration of this manuscript.

Yours sincerely,

Kana Yamamoto

Medical Governance Research Institute

2-12-13-201, Takanawa

Minato, Tokyo 108-0074,

Japan

E-mail: kanachan.y.0508@gmail.com

Tel: +81-3-6455-7401

Fax: +81-3-3441-7505

RESPONSE TO REVIEWERS

Thank you for your detailed review of the content of our manuscript. Responses to your comments are given below. We have revised our manuscript based on your comments and suggestions.

Reviewer 1

This communication aimed to examine the effectiveness of the SARS-Cov_2 booster administration among Japanese adolescents. The topic is very important; however, I have several concerns.

Comment 1: First, I am confused about the differences between the effectiveness of boosters and the effectiveness of booster administration. As I read the title, I think booster administration may indicate the level of adolescents to have a booster, but it was actually to examine the effect of infection. I think the effectiveness of the booster may be more appropriate.

Reply: Thank you for your constructive comment. We modified the title of manuscript to replace the term ‘effectiveness of booster administration’ with ‘effectiveness of booter’.

‘Effectiveness of the booster of SARS-CoV-2 vaccine among Japanese adolescents: A cohort study’ (Lines 2–3)

Comment 2: Second, it will be better if the authors could provide more descriptions of the regression. What is the rationale to use the Poisson regression? Did the data meet the assumption of the Poisson regression?

Reply: We changed to the statistical evaluation with the negative binomial regression since we observed the overdispersion where the variance is greater than mean in our data.

‘We employed the negative-binomial regression model to calculate the crude incidence rate and the 95% confidence interval (CI) with the group-level counted data since the over-dispersion was observed in the data [12,13].’ (Lines 81–83)

Comment 3: Finally, it is good that the authors made several international comparisons. It will better if the authors could introduce Japanese experiences on both primary and booster vaccines to improve adolescent intake.

Reply: We searched articles on the effectiveness of a SARS-CoV-2 vaccine booter in Japanese adolescents; however, no population-level data have been published so far. The drug approval was made based on the international clinical trials. An article on the vaccine effectiveness of a booster in the Japanese adult population was published (Akaishi et al. Sci Rep. 2022;12;13589). We modified the descriptions in the discussion section, as follows;

‘Japan maintained universal health coverage; however, the vaccination and infection records were separately stored by municipalities and by the public health office of the prefectural government, respectively, due to differences in the regulation. Such difference in the responsible body of data storage makes it difficult to scientifically analyze real-world vaccine effectiveness. To our knowledge, no reports on the vaccine effectiveness of a boost-er have been published in the Japanese adolescent population. An adult study in Miyagi Prefecture, Japan, analyzed the vaccine booster effectiveness in the omicron era; however, the study was conducted in the outpatient clinics where the individuals with close contact to COVID-19 patients [22]. Of note, the Miyagi study supports the third dose of the SARS-CoV-2 vaccination to prevent infection as 40% (95%CI: 20–60%) of the effectiveness, which was the highest at 10–70 days after receiving the third dose.’ (Lines 138–149)

We sincerely thank the reviewers’ valuable comments and hope that the revised manuscript would address the comments.

Reviewer 2 Report

There are several issues and methodological shortcomings that dampen my interest in the present communication. The analyses should be re-checked for accuracy.

Specific comments:

1. "... there is, however, growing evidence of long-term, physical and social impacts of COVID-19 on adolescents" - what exactly do you mean? And can vaccination ameliorate these risks? Rather, the author should state that studies in adults suggest that COVID-19 vaccination is associated with a lower risk of several, but not all, potential long-COVID sequelae in those with breakthrough SARS-CoV-2 infections (citation: pubmed.ncbi.nlm.nih.gov/35447302).

2. Vaccine efficacy and effectiveness are not the same thing and should not be conflated as such. Efficacy is the degree to which a vaccine prevents disease, and possibly also transmission, under ideal and controlled circumstances, comparing a vaccinated group with a placebo group. Effectiveness meanwhile refers to how well it performs in the real world. This is an important distinction.

3. "Physicians can diagnose COVID-19 if the individuals without any SARS-CoV-2 testing show cold symptoms and had close contact with known COVID-19 patients" - if I am reading this correctly, this would seriously affect the reliability of the findings as the lack of testing in the first place would reduce pickup rate. What is the general testing and pickup rate for SARS-CoV-2 in the country? At least some comments are necessary.

4. "The biases due to misclassification and prior infection might minimize" - this is not true since the study utilised a very short observation period and only calculated crude incidence rates.

5. At least some comments on vaccine safety would be helpful, especially for this age group, as post-vaccination perimyocarditis continues to be an issue of concern that creates vaccine hesitancy (citation: pubmed.ncbi.nlm.nih.gov/36146535).

Author Response

November 6, 2022

Dr. Prof. Dr. Ralph A. Tripp,

Editor-in-chief,

Vaccines,

Dear Dr. Tripp,

I am writing to submit a revision of our manuscript titled “Effectiveness of the booster of SARS-CoV-2 vaccine among Japanese adolescents: A cohort study” (Manuscript ID: vaccines-1990162). We sincerely appreciate the helpful comments from the reviewers and have revised our manuscript according to their suggestions.

The changes we have made to our manuscript are recorded with the track and change service. Please find below the detailed list of changes made in response to the reviewers' comments, along with an explanation for these changes.

We hope that the revised manuscript will now be acceptable for publication in the Vaccines and look forward to hearing your decision. Thank you very much for your consideration of this manuscript.

Yours sincerely,

Kana Yamamoto

Medical Governance Research Institute

2-12-13-201, Takanawa

Minato, Tokyo 108-0074,

Japan

E-mail: kanachan.y.0508@gmail.com

Tel: +81-3-6455-7401

Fax: +81-3-3441-7505

RESPONSE TO REVIEWERS

Thank you for your detailed review of the content of our manuscript. Responses to your comments are given below. We have revised our manuscript based on your comments and suggestions.

Reviewer 2

Comment 1: "... there is, however, growing evidence of long-term, physical and social impacts of COVID-19 on adolescents" - what exactly do you mean? And can vaccination ameliorate these risks? Rather, the author should state that studies in adults suggest that COVID-19 vaccination is associated with a lower risk of several, but not all, potential long-COVID sequelae in those with breakthrough SARS-CoV-2 infections (citation: pubmed.ncbi.nlm.nih.gov/35447302).

Reply: Thank you for your constructive comments. We re-described the rationale and role of the SARS-CoV-2 vaccination to the adolescents with the reference the reviewer suggested

‘Adolescents tend to exhibit milder symptoms of the coronavirus disease-2019 (COVID-19) compared to adults [1]; however, a surge of COVID-19 infections in adolescents can exacerbate the outbreak to the whole population [2], and the long-COVID sequelae such as tiredness, headache, and shortness of breath are major concerns on the adolescents' health [3,4]. Vaccination is effective in preventing COVID-19-related hospitalization among all age groups, which supports the role of the severe acute respiratory syn-drome coronavirus 2 (SARS-CoV-2) vaccination in adolescents [5]. Of note, the vaccine effectiveness to the long-COVID symptoms in adolescents has not been evident yet [6,7].’ (Line 28–36)

Comment 2: Vaccine efficacy and effectiveness are not the same thing and should not be conflated as such. Efficacy is the degree to which a vaccine prevents disease, and possibly also transmission, under ideal and controlled circumstances, comparing a vaccinated group with a placebo group. Effectiveness meanwhile refers to how well it performs in the real world. This is an important distinction.

Reply: Since we focused on the real-world effectiveness of the booster vaccination of SARS-CoV-2, we excluded the descriptions on the term ‘efficacy’ as the reviewer suggested.

Comment 3: "Physicians can diagnose COVID-19 if the individuals without any SARS-CoV-2 testing show cold symptoms and had close contact with known COVID-19 patients" - if I am reading this correctly, this would seriously affect the reliability of the findings as the lack of testing in the first place would reduce pickup rate. What is the general testing and pickup rate for SARS-CoV-2 in the country? At least some comments are necessary.

Reply: Clinical diagnosis of COVID without the testing is very limited when the patients are difficult to have the SARS-CoV-2 testing. We counted the number of the clinically diagnosed COVID patients without testing in the Fukushima Prefecture during the observation period. We added the following sentences in the discussion section.

‘The clinical diagnosis of COVID-19 without the SARS-CoV-2 testing affects the reliability of the finding of this study; however, the no-test diagnosis was very limited. Twenty-three (0.3%) cases among 7,353 COVID patients in Fukushima Prefecture during the study period were diagnosed as COVID-19 without testing due to difficulty in having the opportunity of the test [39]’ (Line 183–187)

Comment 4: "The biases due to misclassification and prior infection might minimize" - this is not true since the study utilised a very short observation period and only calculated crude incidence rates.

Reply: We deleted the comments and corrected the descriptions as follows;

‘False-negative test results of PCR or antigen tests cause misclassification [36,37]. We could not identify patients infected before the observation period, which can also contribute to bias [38]. The short observation period might influence the biases due to misclassification and prior infection.’ (Line 180–183)

Comment 5: At least some comments on vaccine safety would be helpful, especially for this age group, as post-vaccination perimyocarditis continues to be an issue of concern that creates vaccine hesitancy (citation: pubmed.ncbi.nlm.nih.gov/36146535).

Reply: Thank you for your valuable comments. We added a paragraph on the discussion to ensure the vaccine safety in the adolescents, as follows;

‘Concern about the SARS-CoV-2 vaccination for adolescents includes adverse events such as perimyocarditis [28,29], which might cause vaccine hesitancy [30]. Soma city conducted recipient-oriented safety surveillance after the primary vaccination of the SARS-CoV-2 and reported the results which are publicly available for adolescents [31]. Safety surveillance is commonly designed at the national level to determine the rate but serious events by using a large database [32]. A small study like the survey in Soma City is not enough to find out unusual events. It is, however, beneficial to obtain the trust of residents by keeping transparency on the influence of vaccination in the community. Soma city is the fastest municipality for the coverage of COVID vaccination, based on lessons learned from the disaster of the Great East Japan Earthquake in 2011 [33]. The city office balanced the vaccine policy between boosting vaccination with a community-based approach and ensuring safety by surveying and reporting adverse events, especially for adolescents. The misinformation which may cause vaccine hesitancy has been wildly spread on internet websites [34]. A detailed assessment is necessary to determine the relationship between their perception of vaccination and the characteristics of websites that they commonly refer for acquiring knowledge of vaccination.’ (Line 158–173)

We sincerely thank the reviewers’ valuable comments and hope that revised manuscript would address the comments.

Round 2

Reviewer 2 Report

Thank you for the revisions.

Specific comments:

1. "The community-based mass vaccination of a booster dose has an advantage not only in the rapid coverage of immunization but in estimating its real-world effectiveness" - this does not really make sense as the purpose of a mass vaccination campaign is to mass vaccinate.

2. "... a grant of the collaborative studies" - please provide the actual grant number and details.

Author Response

November 8, 2022

Dr. Prof. Dr. Ralph A. Tripp,

Editor-in-chief,

Vaccines,

Dear Dr. Tripp,

I am writing to submit a revision of our manuscript titled “Effectiveness of the booster of SARS-CoV-2 vaccine among Japanese adolescents: A cohort study” (Manuscript ID: vaccines-1990162). We sincerely appreciate the helpful comments from the reviewer and have revised our manuscript according to their suggestions.

The changes we have made to our manuscript are recorded with the track and change service. Please find below the detailed list of changes made in response to the reviewer’s comments, along with an explanation for these changes.

We hope that the revised manuscript will now be acceptable for publication in the Vaccines and look forward to hearing your decision. Thank you very much for your consideration of this manuscript.

Yours sincerely,

Kana Yamamoto

Medical Governance Research Institute

2-12-13-201, Takanawa

Minato, Tokyo 108-0074,

Japan

E-mail:

kana.yamamoto@megriconnect.net

kanachan.y.0508@gmail.com

Tel: +81-3-6455-7401

Fax: +81-3-3441-7505

RESPONSE TO REVIEWERS

Thank you for your detailed review of the content of our manuscript. Responses to your comments are given below. We have revised our manuscript based on your comments and suggestions.

Reviewer 2

Comment 1: “The community-based mass vaccination of a booster dose has an advantage not only in the rapid coverage of immunization but in estimating its real-world effectiveness” - this does not really make sense as the purpose of a mass vaccination campaign is to mass vaccinate.

Reply: We correct the sentence as follows;

‘The community-based mass vaccination of a booster dose has an advantage not only in the broad coverage of immunization but in estimating its real-world effectiveness.’ (Line 130–131)

Comment 2: “... a grant of the collaborative studies” - please provide the actual grant number and details.

Reply: This is an internal grant of Fukushima Medical University, and no grant number was provided. We put the grant title in the acknowledge section as follows;

‘This study is supported in part by a grant of the collaborative studies for radiation disasters and medical science in the Fukushima Medical University, entitled ‘The long-term health survey for the adolescents evacuated after the Great East Japan Earthquake’ (Fukushima, Japan),’ (Line 214–216)

We sincerely thank you again for the valuable comments and hope that the revised manuscript will address the comments.
